# Recombinant Thaumatin-Like Protein (rTLP) and Chitinase (rCHI) from *Vitis vinifera* as Models for Wine Haze Formation

**DOI:** 10.3390/molecules27196409

**Published:** 2022-09-28

**Authors:** Wendell Albuquerque, Pia Sturm, Quintus Schneider, Parviz Ghezellou, Leif Seidel, Daniel Bakonyi, Frank Will, Bernhard Spengler, Holger Zorn, Martin Gand

**Affiliations:** 1Institute of Food Chemistry and Food Biotechnology, Justus Liebig University Giessen, Heinrich-Buff-Ring 17, 35392 Giessen, Germany; 2Institute of Inorganic and Analytical Chemistry, Justus Liebig University Giessen, Heinrich-Buff-Ring 17, 35392 Giessen, Germany; 3Department of Beverage Research, Geisenheim University, Von-Lade-Straße 1, 65366 Geisenheim, Germany; 4Fraunhofer Institute for Molecular Biology and Applied Ecology, Ohlebergsweg 12, 35392 Giessen, Germany

**Keywords:** thaumatin-like protein, chitinase, haze, sulfite, polyphenols, wine, protein

## Abstract

Cross-linking net aggregates of thermolabile thaumatin-like proteins (TLPs) and chitinases (CHIs) are the primary source of haze in white wines. Although bentonite fining is still routinely used in winemaking, alternative methods to selectively remove haze proteins without affecting wine organoleptic properties are needed. The availability of pure TLPs and CHIs would facilitate the research for the identification of such technological advances. Therefore, we proposed the usage of recombinant TLP (rTLP) and CHI (rCHI), expressed by *Komagataella phaffii*, as haze-protein models, since they showed similar characteristics (aggregation potential, melting point, functionality, glycosylation levels and bentonite adsorption) to the native-haze proteins from *Vitis vinifera*. Hence, rTLP and rCHI can be applied to study haze formation mechanisms on a molecular level and to explore alternative fining methods by screening proteolytic enzymes and ideal adsorptive resins.

## 1. Introduction

Wine haze is generated by insoluble protein aggregates large enough to scatter light and lead to a loss of transparency. Such protein flocculation result from the thermolabile grape pathogenesis-related (PR) proteins or heat-unstable proteins (HUPs), predominantly thaumatin-like proteins (TLPs) and chitinases (CHIs). Moreover, these protein-protein interactions are influenced by numerous wine matrix components such as polyphenols [1], metal and sulfite ions [2], organic acids [3] as well as specific physicochemical conditions such as moderate temperature [4,5], high ionic strength [6] and acidic pH [7]. Although TLPs are reported to have hydrophobic spots that bind polyphenolic compounds [8] and irregular structures that interact via intermolecular disulfide bridges [9,10], CHIs are reported to be less stable and to denature irreversibly [11]. In addition, minor variations in pH conditions and ionic strength can affect the interaction of polyphenols with hydrophobic residues of HUPs, since their conformational states depend on their isoelectric points (pI) and the net charges that are present on the protein surfaces [12]. Therefore, the understanding of the interaction between thermolabile wine proteins and other wine matrix components is a fundamental step in the search for novel clarification methods [9].

Since haze in non-fined wines is mainly induced by temperature variations during transportation and storage, it eventually causes consumer aversion due to an unpleasant appearance [5,13]. To avoid this, high amounts of bentonite clay are still applied in clarification processes, although its adsorptive cationic character has deleterious effects on wine aroma, taste, and volume [14,15]. Due to its economic impact, winemakers have considered replacing bentonite with alternative cost-effective fining methods, demanding research and technological advances. However, isolation of purified wine haze protein fractions on a technical scale is too cumbersome to become a commonly applied method. The “large-scale” availability of pure TLP and CHI would facilitate the screening of novel fining agents for the selective removal of HUPs and prevent detrimental effects on other wine matrix components.

We proposed the comparison of recombinant TLPs and CHIs (rTLP and rCHI) to serve as alternative models of wine haze proteins. Therefore, those proteins were heterologously expressed by *Komagataella phaffii* and compared with their corresponding native-host proteins in terms of glycosylation, melting point, aggregation potential, adsorption by bentonite and functionality (CHI activity). Moreover, we studied the recombinant proteins as models for their haze potential influenced by haze inducers such as polyphenols and sulfite ions employing heat tests [16,17,18]. Hitherto, such tests have been applied in experiments that studied haze formation using HUPs directly isolated from *V. vinifera* under the influence of other wine matrix components [1,19]. The use of rCHI and rTLP may be crucial for the search of proteolytic enzymes and adsorptive agents as alternative fining agents, providing the opportunity to find suitable and profitable alternatives for winemakers. With such recombinant haze models, a better understanding of haze mechanisms and future alternative strategies to bentonite fining, such as proteolytic treatments, can be conveniently achieved and aid in resolving the “haze challenge” in the wine industry.

## 2. Results

### 2.1. Validation of Transformants

After plasmid isolation from Escherichia coli cells, the correctness of the plasmids was confirmed by DNA sequencing (Appendix A) and by agarose gel electrophoresis. The DNA excised from gel bands at the molecular size of approximately 8000 bp and 3000 bp (BglII digestion) and 10,000 bp (SacI digestion) was integrated into the *K. phaffii* genome (Appendix A). Afterward, the transformed *K. phaffii* cells were grown on a dextrose medium deficient in histidine (MD-His) and geneticin agar plates (Appendix A, respectively). All *K. phaffii* transformants were considered to have a Mut^+^ phenotype as they grew in both MD and MM media (Appendix A) and harbor an intact AOX1 gene (Appendix A).

### 2.2. Heterologous Expression of rTLP and rCHI

In total, 80 clones (exemplified in Appendix A), 20 for each of the four electroporation batches (rTLP plasmid digested by BglII (1), and by SacI (2), rCHI plasmid digested by BglII (3), and by *Sac*I (4) (see Appendix A) from the transformations (Appendix A), were screened. The clones with the highest protein expression levels (based on the protein band intensities on WB membranes) were selected for further analysis (9 clones for rTLP and 6 for rCHI). SDS-PAGE analysis of the selected clones showed dense bands of expressed protein with approximately 23 kDa (Figure 1a) for rTLP, and a double protein band at 27–32 kDa representing rCHI (Figure 1a,b). The correlation between the visualized bands, the clones, and the respective electroporation batches is shown in Appendix A. Protein bands representing rTLP and rCHI were further identified by WB (Figure 1b).

### 2.3. Purification of rTLP and rCHI and Characterization by MS-Based Bottom-Up Proteomics

IMAC chromatograms exhibited a major peak of proteins eluted by an imidazole-containing buffer (Figure 1c) for both rTLP (Appendix A) and rCHI (Appendix A). Their purification was confirmed by SDS-PAGE and WB, as shown in Figure 1c. A further SEC [calibrated with protein standards (Appendix A)] purification provided a peak with a retention time corresponding to molecular masses of 20 to 40 kDa (Appendix A) for both recombinant proteins, which was also confirmed by SDS-PAGE and WB (Figure 1d).

The LC-MS/MS analysis of the digested protein bands confirmed the expression of rTLP (5 unique peptides) and rCHI (8 unique peptides) confidently. The identified peptides and their locations of conformity in the corresponding sequences are presented in Figure 1e, Appendix A.

### 2.4. TLP and CHI: Recombinant Versus Native Proteins

#### 2.4.1. Glycosylation Analysis

Figure 2a,b show the comparison of the recombinant proteins (rTLP and rCHI), both treated (for cleavage of attached *N*-glycans) and not treated with PNGase Endo H. The bands observed corresponded to rTLP (24 kDa), rCHI (27 kDa) and PNGase (34 kDa). No significant MW shifts were observed for rTLP. However, for rCHI, other protein bands were observed between 20 and 27 kDa after treatment by PNGase Endo H (Figure 2b). Figure 2c shows the separation of rTLP and rCHI in an electrophoresis gel after Schiff staining. A dense band corresponding to the highly glycosylated mucin protein was observed at 200 kDa, and a faint protein band compared to rCHI was detected at about 27–32 kDa. The computational analysis (performed by NetNGlyc v.1 [20] and NetOGlyc v.4.0 [21] of potential glycosylation sites is shown in Figure 3d,e. Moreover, two glycans [(Xyl)_1_-(GlcNAc)_3_-(Man)_4_ and (Man)_3_] attached to a tryptic peptide (KDYCSQLGVSPGDNLTC) of rCHI were detected by LC-MS/MS analysis (see Figure 3f and Appendix A). No glycans attached to peptides from rTLP were detected in the MS-based analysis.

#### 2.4.2. Chitinolytic Activity

The chitinolytic activity of rCHI was evidenced by visualization of a degradation halo of chitin (Appendix A) embedded in an agarose gel, as shown in Figure 2d. The radii of the halos (proportional to the enzyme activity) were 1.5 cm for rCHI, 1 cm for cCHI and 0.9 cm for the SF wine protein samples (Figure 2g and Appendix A). The denatured cCHI (marked as a control in Figure 2d) did not show any degradation zone. The quantitative analysis of the chitinolytic activities by DNS assays is shown in Figure 2h. For both chitosan and chitin substrates, high levels of rCHI activity against the substrates were observed, and the activities were similar to those of CHI from *S. griseus* and of CHIs present in the SF wine. When cCHI was heat-denatured, its activity was strongly reduced or almost completely abolished (Figure 2h).

#### 2.4.3. Thermostability of rTLP and rCHI

The melting curves of rTLP and rCHI based on the CPM fluorescence signal are shown in Figure 2i,j, respectively. For both proteins, the fluorescence started to increase (protein melting) at approximately 55 °C. The curve for rTLP (Figure 2i) presented a sigmoidal form with an exponential phase from 58 to 68 °C, reaching a plateau afterwards. The sigmoidal-shaped curve of rCHI (Figure 2j) started its exponential phase at about 54 °C and reached its steady state at 63 °C. The first derivative of each melting curve (Figure 2i,j) defined melting temperatures of 63 °C and 59 °C for rTLP and rCHI, respectively.

#### 2.4.4. Adsorption of the Proteins to Bentonite (Bentonite Fining)

Both rTLP and rCHI did not form haze under a heat test after treatment with bentonite at final concentrations of 0.25, 0.5 and 1 g/L (Figure 2k). Likewise, proteins from the SF wine could also be fined by the three different bentonite concentrations tested (Figure 2k). The correlation between different concentrations of the applied bentonite (0.5, 0.25, 0.125, 0.05, 0.02 g/L) and the residual haze formed is shown in the Figure 2l and Appendix A. Haze levels increased following a decrease in the bentonite concentration, more distinctly from 0.25 g/L to 0.02 g/L.

#### 2.4.5. Influence of Polyphenols and Sulfite Ions on the Haze Potential of rTLP and rCHI

Figure 3 correlates the aggregation potential of rTLP and rCHI to the haze levels (at 540 nm) of the respective solutions (and the residual pellet formed) after a heat test. Both rTLP and rCHI (at 0.25 mg/mL) formed hazy solutions and residual protein pellets after a heat test (Figure 3a,b), even in the absence of haze inducers. This effect was enhanced if both proteins were used (Figure 3c), although the same solutions containing pure gliadins (negative control) did not form haze (Figure 3d). The SF colloids (Figure 3e) formed haze similarly to rTLP and rCHI, with protein aggregation highly induced by sulfite ions (at 1 mg/mL) and polyphenol extracts (at concentrations of 0.25 mg/mL and 0.5 mg/mL).

The identified polyphenolic compounds (Appendix A and Appendix A) in the extracts were: caffeic acid [2.9 mg/g (milligram per gram of the grape juice dry extract)], caftaric acid (3.9 mg/g), catechin (6.1 mg/g), coutaric acid (4.9 mg/g), epicatechin (10.7 mg/g), fertaric acid (1.1 mg/g), grape reaction product (GRP, 2-*S*-glutathionyl caftaric acid) (1.6 mg/g), *p-*coumaroyl–glucosyl–tartrate (*p*-CGT) (0.5 mg/g), procyanidin B1 (1.5 mg/g), procyanidin B2 (6.3 mg/g), procyanidin C1 (6.7 mg/g), protocatechuic acid (1.6 mg/g) and quercetin-3-*O*-glucoside (Que-3-glc) (0.7 mg/g). Moreover, the monosaccharide content was 17.97% (Appendix A). In particular, adding the polyphenol extract to rTLP and rCHI model solutions promoted a two-fold increase in haze formation (compared to the samples in the absence of polyphenols). Under such conditions, the total haze formed was slightly higher for rCHI than for rTLP, reaching differences between them of about 21.7% and 5.4%, at polyphenol concentrations of 0.25 and 0.5 mg/mL, respectively (Figure 3e). When both proteins were combined (rTLP + rCHI) haze levels were strongly enhanced, up to 37.5% (at a polyphenol concentration of 0.25 mg/mL) and 78.4% (at 0.5 mg/mL), as shown in Figure 3c,f. The protein aggregates were visualized as a pellet after centrifugation (Figure 3 and Appendix A).

The haze levels of model solutions of rTLP and rCHI increased at higher concentrations of sulfite ion (1 mg/mL). At the same time, the absorbances were comparable to or slightly lower than those of the pure proteins without additives at 0.5 mg/mL (Figure 3f). At high sulfite concentrations, samples containing rTLP were 8.2% more turbid than those containing rCHI. The combination of two HUP species (rTLP + rCHI) was not crucial to induce higher haze levels under the influence of sulfite ions.

## 3. Discussion

### 3.1. Molecular Characterization and Comparison with Native Proteins

Transformants of *K. phaffii* (GS115) successfully expressed high levels of thermolabile rTLP and rCHI. Using a eukaryotic host organism provided the cell machinery to obtain recombinant HUPs similar to the native ones from *V. vinifera* in terms of glycosylation, melting point, and functionality (enzymatic activity). Furthermore, their ability to aggregate in acidic solutions (pH 4) demonstrated their potential to be regarded as haze-forming model proteins. Such histidine-tagged haze proteins have the advantage that they can be produced on a large scale and may be easily purified.

Grape HUPs play an essential role in fungal defense processes, which are regulated via post-translational modifications [22,23] including potential glycosylation reactions, which was reported by Palmisano et al. [24]. Using tandem MS analysis, the authors identified glycopeptides belonging to a putative thaumatin-like protein (accession: gi|7406714, from *V. vinifera*), a class IV chitinase (accession: gi|164699029, *Vitis pseudoreticulata*) and a class IV endochitinase [accession: gi|2306813, *V. vinifera*) from a Chardonnay white wine. In our experiments, the identified double “his-tagged” protein bands in the purified rCHI (SDS-PAGE) indicated a partial glycosylation (also detected by the Schiff method). In addition, the presence of other protein bands with a lower molecular mass after incubation with the enzyme PNGase suggests the presence of glycosylated, non-glycosylated and partially deglycosylated forms. LC-MS/MS analysis of the “his-tagged” double bands of rCHI (about 27–30 kDa) also confirmed that both bands are related to class IV chitinases. Furthermore, the peptide with amino acid sequence of KDYCSQLGVSPGDNLTC from rCHI (the one with a potential *N*-glycosylation site) was also experimentally found to be glycosylated, which was confirmed by the presence of different glycans attached to the Asn (N) residue. What proves the capability of *K. phaffii* to provide similar glycosylation levels of rCHI as *V. vinifera*. This is crucial as differences in the glycosylation level between native and heterologous HUPs would influence their functionality, since glycans (part of *N*-glycosylated residues) can interfere with protein folding [25] and inhibit protein aggregation, thermolysis and proteolysis [24]. The glycosylation analysis using NetNGlyc and NetOGlyc showed that both proteins are putatively glycosylated, and up to five potential glycosylation sites in TLP and CHI were observed. For a putative TLP from a Chardonnay white wine, Palmisano et al., [24] identified the Asn134 residue as being glycosylated. Asn134 of rTLP is followed by Pro135 and Thr135, and *N*-glycosylation is probably hindered by the presence of the close proline residue of the recombinant protein. Additionally, three putative *O*-glycosylation sites are located at the residues Thr170, Thr171 and Thr193 (Figure 2d).

For rCHI, an *N*-glycosylation site was identified at Arg261 (experimentally confirmed in the rCHI and also mass spectrometrically identified by Palmisano et al. [24]) and four *O*-glycosylation sites at the residues Ser55, Ser56, Ser57 and Ser62 (Figure 2e). Landim et al. [26] reported *O-*glycosylation in a class I chitinase belonging to the glycoside hydrolase family 19 with one *N*-terminal carbohydrate-binding module (CBM) of the family 18 from the plant *Vigna unguiculata* expressed in *K. phaffii*. However, plant class IV CHIs, to which the rCHI reported in this study also belongs to, are classified as proteins of the glycoside hydrolase family 19 with one *N*-terminal CBM of the family 18. They are also reported to have shorter sequences (also fewer subsides at the catalytic cleft) than other CHI classes [27]. Similar to the class I CHI of *V. unguiculata*, the functionality of the rCHI was confirmed by its potential to degrade chitinous substrates. In our case, the acid-hydrolyzed chitin was embedded in agar, comparable with the chitinolytic activity of chitinases present in an SF colloid (Figure 2g). Moreover, rCHI exhibited high chitinolytic activity, similar to that of a commercial CHI in the DNS assays, confirming a proper folding, which is necessary for the enzymatic activity.

The CPM fluorogenic dye binds to free and exposed sulfhydryl groups, revealed under protein denaturation and, therefore, can be applied in thermofluor assays [28]. According to Eilers et al. [29], Cys residues mediate helix interactions (they are often located at helix-helix interaction sites) and they can work as sensors for protein denaturation [30]. Cys residues of wine haze proteins are exposed during protein denaturation and can cross-link proteins under *S*-sulfonation [9,31]. The increase in the CPM fluorescence signal under gradual denaturation of rTLP and rCHI evidences the exposure of Cys residues, which can participate in protein S-S exchanges from temperatures above 55 °C, rearranging disulfide bridges along the polypeptide chains and consequently promoting aggregation [9]. In the thermal shift assays, rCHI showed a lower T_m_ (59 °C) than rTLP (63 °C). This has already been reported by Falconer et al. [11], who found a T_m_ of approximately 55 °C for CHI and a T_m_ of approximately 62 °C for TLP isoforms from a Sauvignon blanc wine, by performing differential scanning calorimetry (DSC). These findings underline the active folding of both recombinantly produced proteins. The CHI structure is reported to have distinctly higher amounts of helices than TLP. About 65% of CHI’s structure is composed of α-helices, whereas TLP has 31% of helical secondary structures [7]. CHI and TLP contain 15 and 16 Cys residues in their polypeptide chains, respectively, which are mostly located outside of helices [32], and could participate in helix-helix interactions [30]. The reduction of the fluorescence signal (from about 65 to 70 °C) for rTLP and rCHI illustrates the quenching of the CPM dye fluorescence caused by protein aggregation and supports the fact that removal of HUPs by proteolytic treatments requires temperatures above 70 °C, as in the case of heat tests [17] or flash pasteurization [33].

rTLP, rCHI and SF colloids could be adsorbed by bentonite clay in concentrations routinely used in winemaking [34]. A bentonite concentration of 0.25 g/L was detected as a threshold value for the loss of the capability to prevent haze formation for the SF wine colloids (probably caused by the presence of other wine matrix components). Different concentrations of alternative adsorptive compounds or resins could be evaluated to bind selectively to rTLP and rCHI in the future. Fining agents such as casein, egg albumin, chitosan and polyvinylpolypyrrolidone (PVPP) were already tested as alternatives to bentonite clay [35], but other agents such as synthetic polymers have still not been well studied [36]. Recently, Sommer and Tondini [37] applied different potential fining agents such as saccharomyces paradoxus (with high concentrations of chitin in its cell wall), polystyrene, chitosan and carboxymethyl cellulose (CMC) to remove wine proteins with low levels of instability.

### 3.2. Influence of Polyphenols and Sulfite Ion on the Haze Potential of TLP and CHI

Both class IV CHI and the TLP isoform 4JRU from *V. vinifera* undergo irreversible denaturation processes, leading to conformational changes, which expose specific amino acids that bind to other polypeptide chains and form aggregates [38]. None of the rTLP or rCHI individually induced more haze than their combination in the presence of polyphenols. The assumption that wine haze is triggered by different classes of proteins was also discussed by Esteruelas et al. [39], when they found a protein mixture in the precipitates formed after heating Sauvignon Blanc wines.

As previously reported in the literature [10], polyphenols induced aggregation of rTLP and rCHI under heating (Figure 4), confirming their role as main haze factors. The concentrations of polyphenols differ in white wines depending on the grape variety and vintage [40] in a range between 220 and 500 mg/L [41]. The cross-linking net formed as a result of the interaction between the polyphenols and the proteins could be visually confirmed by the color of the protein aggregates (Figure 4). In theory, hydrophobic residues in the protein backbone, which are exposed after a denaturation process, can interact with polyphenols through various reactions [9,41]. According to Pocock et al. [19], gallic acid and caffeic acid are haze inducers, although caftaric acid, epicatechin and ferulic acid did not affect haze levels in model solutions. In addition, putative binding sites in TLP have already been identified for quercetin and caffeic acid [42]. Marangon et al. [8] showed the potential of tannins from a Pinot Grigio wine to promote protein aggregation. The authors discussed that the conformational mobility of phenolic molecules seems to be essential for the polyphenol-protein binding associated with stacking-stacking interaction between the planar proline residues in proteins and the phenolic rings [14,43].

Sulfur dioxide is a common additive in winemaking, and it is normally concentrated between 0.05 and 1.8 mg/mL [19]. These authors added different concentrations (0, 0.5, 1, 1.5 and 2 mg/mL) of potassium hydrogen sulfate to model wines containing purified HUPs and observed a concentration of 1 mg/mL as crucial to developing haze. In our experiments, sulfite ions showed a secondary role in haze formation. For the induction of protein aggregation, high concentrations of sulfite ions were required. Moreover, haze levels also did not increase in the presence of sulfite ions when the recombinant proteins were combined (rTLP + rCHI).

Chagas et al. [2], Marangon et al. [6] and Pocock et al. [19] observed that the turbidity of wine model solutions increased proportionally to the protein and sulfite concentrations. That was explained by a rearrangement of disulfide bonds between proteins caused by sulfonation reactions that eventually resulted in aggregation. The authors comparatively verified that the TLP isoform 4JRU from *V. vinifera* aggregates more in the presence of sulfite than the class IV CHI.

### 3.3. Heterologous rTLP and rCHI as Haze-Forming Protein Models for Research and Applications

Recently, many technological advances for the analysis of wine haze proteins have been made [44] and they are still in demand to facilitate the research of adapted clarification processes. Since pure recombinant haze proteins can be produced large scale, their use has a high potential to evaluate various fining agents such as adsorptive compounds [37], resins [45,46], and peptidases [32]. Nano (magnetic) particles with functionalized surfaces have been recently developed for clarification purposes [47]. Acrylic-acid plasma-polymer-coated magnetic nanoparticles (AcrAppMNP) have been applied for the removal of TLPs and CHIs from wines in a fast method and without affecting organoleptic properties [44]. Yang et al. [48] confirmed the haze potential of recombinant GRIP32 as a novel haze protein in wines and the influence of polyphenols as haze inducers. The authors demonstrated that procyanidins (PCs) or epigallocatechin gallate (EGCG) interacted with the GRIP32 proteins to form aggregates and studied the roles of polysaccharides in hindering protein-protein interactions.

The recent application of top-down proteomics as an advanced MS technique showed a promising strategy for characterizing intact wine proteins [32], which can be applied to rTLP and rCHI to locate cleavage spots by various peptidases comprehensively. The use of recombinant proteins in the study of haze formation offers enormous possibilities to study novel clarification methods, especially by applying agents that offer high affinities to bind both TLP and CHI, preventing the concomitant removal of polyphenolic compounds or organic acids that are deemed essential for the taste of wine. The screening for peptidases, which possess the ability to cleave TLP and CHI in acidic pH and low temperatures specifically, would be facilitated by the improved availability of pure haze proteins. Alternatively, rTLP and rCHI could be used as experimental models to test different variations of pH, temperature, ionic strength, and different concentrations of wine matrix components (polysaccharides, sulfite, and polyphenols) aiming to find conditions of high protein aggregation, which should be avoided in real winemaking processes.

## 4. Materials and Methods

### 4.1. Plasmid Amplification and Isolation

The codon-optimized genes for a TLP (UniProt ID: F6HUG9) and a CHI (UniProt ID: Q7XAU6) with nucleotides coding for a hexahistidin-tagged at the 5’-end, encoding a thermo-labile thaumatin-like protein isoform (PDB code 4JRU) and a class IV chitinase from *V. vinifera*, respectively, were synthesized by BioCat GmbH (Heidelberg, Germany) and cloned into a pPIC9K vector [with *EcoR*I/*Not*I restriction sites (plasmid vectors are shown in Appendix A)]. The plasmids were transformed into *Escherichia coli* competent cells (NEB^®^ 10-beta cells, New England Biolabs, Ipswich, MA, USA), cultured in 5 mL of Lysogeny Broth (LB)-ampicillin medium (Appendix A) with a working concentration of 100 µg/mL (overnight at 37 °C and 180 rpm), followed by the main culture in 100 mL of LB-Amp (ampicillin) medium (for 6 h at 180 rpm and 37 °C). After harvesting cells by centrifugation (4000× *g*, 20 min, 4 °C), the plasmids were subsequently: (1) isolated with a PureLink™ HiPure Midiprep kit (Thermo Fisher Scientific, Waltham, MA, USA); (2) sequenced by Sanger sequencing (Eurofins Genomics Germany GmbH, Ebersberg, Germany); (3) linearized by restriction enzymes (*Bgl*II and *Sac*I both from Thermo Fisher Scientific); and (4) separated by 1% agarose gel electrophoresis. Plasmid fragments (excised from gels) were dissolved in a binding buffer (NTI buffer) and the DNA was extracted with a NucleoSpin^®^ Gel (PCR Clean-up) Kit (Macherey-Nagel GmbH & Co. KG, Düren, Germany), and quantified by nanophotometry at 260/280 nm (Pearl spectrophotometer, Implen GmbH, Munich, Germany).

### 4.2. Transformation into K. phaffii, Selection of Transformed Cells and Phenotype Determination

Linearized DNA plasmids were transformed into the *K. phaffii* strain GS115 (Invitrogen AG, Carlsbad, CA, USA) by electroporation. For that, yeast cells were: (1) cultivated in 100 mL of yeast-peptone-dextrose (YPD) liquid medium (at 30 °C and 200 rpm, until the OD_600_ reached 1.2); (2) harvested by centrifugation (4000× *g*, 20 min, 4 °C); (3) resuspended in 250 mL ice-cold water; (4) centrifuged again (4000× *g*, 20 min, 4 °C); and (5) finally resuspended in 1 mL of ice-cold 1 M sorbitol. Electroporation was performed by adding 10 µL (1 µg/µL) of the linearized DNA plasmid solution to 80 µL of the cell suspensions in chilled sterile cuvettes in which an electroporator (Eporator^®^, Eppendorf SE, Hamburg, Germany) was inserted following standard protocols [49].

Transformants were screened by cultivation on his-deficient selective agar plates (his-selective medium, at 30 °C for 72 h) and by their antibiotic resistance, culturing them on geneticin agar plates (at 30 °C for 96 h). For stable storage, the clones were grown on YPD agar (see Appendix A) plates and stored at 4 °C.

For phenotype determination, transformants were cultured (at 30 °C for 48 h on methanol minimal medium (MM) and dextrose minimal medium (MD) agar plates (Appendix A) for selecting phenotypes Mut^+^ (active *AOX1* and *AOX2* genes) or Mut^S^ (p*AOX1* gene knocked out). The insertion of the gene p*AOX*1 was further evaluated by PCR amplification by lysing the transformants and following standard PCR procedures, using the primers Pichia_*AOX1*_fw and Pichia_*AOX1*_rv.

### 4.3. Recombinant Expression, Purification and Identification of rTLP and rCHI

#### 4.3.1. Recombinant Expression

Transformants were picked from stocks and inoculated in 50 mL (in 250 mL baffled Erlenmeyer flasks) buffered glycerol-complex medium (BMGY) (at 30 °C and 200 rpm, for 24 h). Subsequently, 1 mL of the BMGY cultures was used to inoculate 100 mL (in 500 mL baffled flasks) of buffered methanol-complex medium (BMMY) (at 30 °C and 200 rpm, for 3 days), with a daily feeding of 1% methanol. Cultures were further scaled up to 600 mL (in 2 L-Erlenmeyer flasks) with an initial inoculum volume of 4 mL (from BMGY cultures). Cells were separated by centrifugation (4000× *g*, 20 min, 4 °C) and the culture supernatant was concentrated by pressure-operated dialysis (Vivaflow ultrafiltration, Sartorius AG, Göttingen, Germany) with molecular mass cut-off (MWCO) of 10 kDa.

#### 4.3.2. Protein Purification

His-tagged proteins were purified through immobilized metal affinity chromatography (IMAC) by using a HiTrap™ IMAC FF (5 mL, Cytiva Europe GmbH, Freiburg, Germany) column coupled to an FPLC system (NGCTM chromatography system, Bio-Rad Laboratories, Munich, Germany). Proteins were eluted at a flow rate of 1 mL/min using a his-elution buffer containing 250 mM imidazole (Carl Roth KG, Karlsruhe, Germany) at pH 7 (Appendix A). The eluted proteins were concentrated and desalted by centrifugal filters with a 10 kDa MWCO (Merck KGaA, Darmstadt, Germany). A second purification step was performed by size exclusion chromatography (SEC) with a HiLoad 16/600 Superdex 75 column (Cytiva Europe GmbH, Freiburg, Germany) with a flow rate of 1 mL/min using 0.1 M TRIS-HCl buffer (pH 7) as mobile phase. Proteins were quantified according to Bradford [50], separated by 12% SDS-PAGE [51] under denaturing conditions, and visualized by Coomassie blue staining. His-tagged proteins were further identified by electroblotting onto a polyvinylidene fluoride (PVDF) membrane through western blot (WB) (Bio-Rad Laboratories). Protein detection was performed by incubation with a primary 6×-his tag monoclonal antibody (HIS.H8) (Thermo Fisher Scientific, Bremen, Germany) and a secondary anti-mouse IgG antibody horseradish-peroxidase (HRP) conjugate (Thermo Fisher Scientific, Bremen, Germany). The bands of the his-tagged proteins were revealed by using an Opti-Dilut 4CNTM substrate kit (Bio-Rad Laboratories).

#### 4.3.3. MS-Based Proteomics Analysis

The recombinant proteins were verified by MS-based proteomics following sample preparation described in Appendix A. The peptides were separated using a Kinetex C18 (2.1 × 100 mm, 2.6 µm, 100 Å, Phenomenex, Torrance, CA, USA) column through an ultra-high-pressure liquid chromatography (UHPLC) system (Dionex UltiMate 3000 RSL, Thermo Fisher Scientific, Bremen, Germany) coupled to a Q Exactive HF-X mass spectrometer (Thermo Fisher Scientific, Bremen, Germany) with instrumental parameters as described by Ghezellou et al. [52]. The recorded raw files were searched against the UniProt database, taxonomically set to *Vitis vinifera*, using Proteome Discoverer software version 2.5 (Thermo Fisher Scientific, Bremen, Germany). The parameters were set to two maximum missed cleavage sites of trypsin digestion, minimum peptide length of 6, MS1 and MS2 tolerances of 10 ppm and 0.5 Da, respectively. The dynamic modification was set to oxidation (+15.995 Da [M]) and static modification to carbamidomethyl (+57.021 Da [C]). Percolator node was used to validate identified peptide-spectrum matches (PSMs) and filter the data with parameters of a strict target FDR (false discovery rate) of 0.01 and a relaxed target FDR of 0.05. The MaxQuant contaminant database was used to mark contaminants in the results file. The MS data are deposited to the ProteomeXchange Consortium via the PRIDE partner repository [53] with the dataset identifier PXD035796 and https://doi.org//10.6019/PXD035796 (accessed on 4 August 2022).

### 4.4. Protein Glycosylation

For removal of possibly-attached glycans, rTLP and rCHI were incubated overnight at 37 °C with PNGase endo H (New England Biolabs), and the molecular masses were compared with those of the non-treated proteins by SDS-PAGE under denaturing conditions. Additionally, the periodic acid-Schiff staining [54] method was performed to detect glycoproteins. The highly glycosylated porcine mucin protein (MW about 200 kDa, Carl Roth KG) was used as a control. In addition, LC-MS/MS data recorded from in-gel digestion of the purified rTLP and rCHI (gel proteins bands) samples were analyzed by the software SimGlycan (Premier Biosoft, Palo Alto, CA, USA) [55] for glycan modifications based on an internal database of carbohydrates.

### 4.5. Chitinolytic Activity

The chitinolytic activity of rCHI was evaluated by degrading chitin from crab shells (Carl Roth KG), which were acid-hydrolyzed as described before [56], and then embedded in agarose gels (agar diffusion method) as described by Zou et al. [57] with minor modifications (Appendix A). Circular holes (of approximately 1 cm radius) were made in the gel and filled with 20 μL of the following protein (0.5 mg/mL) samples: (a) purified rCHI; (b) native (non-heated) CHI from *Streptomyces griseus* (cCHI, a commercial chitinase, which was used as positive control, Merck KGaA); (c) denatured (heated at 80 °C) CHI from *S. griseus* (0.5 mg/mL) (used as negative control); and (d) lyophilized protein fractions of a Silvaner Franken (SF) wine (in 0.1 M Tris-HCl buffer pH 7) (used as a control). The samples were incubated overnight at 37 °C. Afterward, chitin was stained by incubation with calcofluor white stain (0.1 g/mL, Merck KGaA) for 10 min, washed with distilled water and left to rest for 1 h at room temperature. In addition, a quantitative estimation of the chitinolytic activity was performed as described by Breuil & Saddler [58] and Brandt et al. [59] by DNS (3,5-dinitrosalicylic acid) assays (Appendix A). All assays were performed in triplicate and a standard curve was established by using different concentrations of *N*-acetylglucosamine (Merck KGaA).

### 4.6. Analysis of Protein Thermostability by Differential Scanning Fluorimetry (DSF)

Protein thermostability was evaluated by differential scanning fluorimetry (DSF) performed according to Alexandrov et al. [30] and Wang et al. [60], in a real-time PCR device (CFX96 system, Bio-Rad Laboratories). For this purpose, 1 μL of a 50 mM solution of the fluorogenic dye CPM [7-diethylamino-3-(4-maleimidophenyl)-4-methylcoumarin] (Merck KGaA) was mixed with 30 μL of the purified protein solutions (about 50 μg/mL of rTLP or rCHI) in a sealed 96-well PCR plate (Bio-Rad Laboratories). Both, the dye and proteins, were solved in dimethyl sulfoxide (DMSO, Carl Roth) concentrated at 10% [61]. A gradient of temperature from 45 °C to 75 °C was set, and measurements were performed at each 0.5 °C with fluorescence excitation and emission wavelengths set at 387 nm and 463 nm, respectively.

### 4.7. Bentonite Fining

rTLP, rCHI and proteins from a SF wine were bentonite-fined according to Pocock et al. [34] and Pocock and Waters [17]. For that, bentonite powder (Merck K*G*aA) was dissolved (5%, *w*/*v*) in pre-heated distilled water. The stock solution was further diluted to three different concentrations (0.25, 0.5 and 1 g/L) and volumes of 0.8 mL were aliquoted and mixed with 0.2 mL of the rTLP or rCHI solutions (at 0.5 mg/mL). The samples were subsequently left to rest at room temperature for 2 h, centrifuged (at 1500× *g* for 20 min) and the supernatants were submitted to a heat test, as described in Section 4.8.1. To measure the adsorption of the samples to the bentonite quantitatively, a correlation between different final bentonite concentrations (0.5, 0.25, 0.125, 0.05 and 0.02 g/L) and the residual haze (after a heat test) was made by performing heat tests with buffered solutions (0.1 M citrate buffer, pH 4) of the rTLP, rCHI and the SF colloids.

### 4.8. Influence of Haze-Inducing Agents on Protein Aggregation

#### 4.8.1. Haze Test

Heat tests were performed according to Pocock and Waters [17]. rTLP, rCHI, gliadin and proteins from a SF wine in buffered solutions (Section 4.8.3, were heated to 75 °C in a heater device HLC (DITABIS AG, Pforzheim, Germany) for 20 min, followed by cooling down to 25 °C. The samples were pipetted into 96 well plates (150 μL per well) and the absorbance values were measured at 540 nm in a microplate reader (Agilent Biotek, Winooski, VT, USA).

#### 4.8.2. Extraction and Analysis of Polyphenols and Monosaccharides from Grape (*V. vinifera*) Juices

Polyphenols from a white grape juice (Niehoffs-Vaihinger Fruchtsaft GmbH, Lauterecken, Germany, 2.5 L) were eluted from a chromatography column [250 mL, Amberlite XAD16 adsorber resin (Merck KGaA) pre-washed with water (5 L)] by adding 250 mL methanol [subsequently removed by evaporation (rotary evaporator, Heidolph Instruments, Schwabach, Germany]. The polyphenol extract was dried by lyophilization for further utilization. The dry extract was dissolved (2 mg/mL) in water (with 1% methanol) and analyzed using a Luna 3u C18(2) column (Phenomenex) through a UHPLC system (Thermo Scientific Ultimate 3000) with and a low-pressure gradient of water with acetic acid and acetonitrile with water and acetic acid (Appendix A) at a flow rate of 0.25 mL/min at 40 °C. Chromatograms were recorded at 280, 320 and 360 nm and visualized by the software ChromeleonTM (Thermo Scientific). Monosaccharides were determined after hydrolysis of 10 mg of the extract by 125 µL of sulfuric acid (72%, Carl Roth) and 1.35 mL water at 120 °C for 1 h. The samples were filled up to 50 mL water and subsequently filtered (0.2 µm pore size, PSE, Avantor Inc., Darmstadt, Germany). Monosaccharides were analyzed via HPAEC-PAD with a Carbo Pac PA-100 column (250 mm × 4 mm, Thermo Fisher) coupled to a Dionex Bio-LC system. The separation of neutral and acid sugars followed different protocols, as described in Appendix A. Calibrations were performed by using polyphenol and monosaccharide standards.

#### 4.8.3. Aggregation Assays

rTLP and rCHI were incubated at 25 °C for 1 h with different concentrations (Figure 4 and Appendix A) of sodium sulfite (Na_2_SO_3_, Carl Roth KG), to obtain its dissociated form ${\mathrm{SO}}_{3}^{2-}$, and polyphenol extracts (both dissolved in 0.1 M citrate buffer pH 4). Aggregation experiments combined 0.25 mg/mL of protein (rTLP, rCHI and gliadin as control) solutions at two different concentrations (1 mg/mL and 0.5 mg/mL, based on Pocock et al. [19]) of sodium sulfite and two concentrations (0.5 and 0.25 mg/mL, based on Gazzola et al. [1]) of polyphenol extracts (Figure 4). Subsequently, haze tests were performed as described in Section 4.8.1 and the absorbance was measured pre (haze) and post (supernatant) centrifugation (12,000× *g* for 10 min) and compared with the visual residual pellets formed. Controls were performed with gliadin (a protein with less haze potential) and solutions without proteins (only polyphenols or sulfite) were used as blanks. The same experiments were performed with lyophilized samples of a SF wine (see Appendix A) to compare the haze potential of recombinant proteins with that of native wine proteins (from *V. vinifera*) under the same experimental conditions. All experiments were performed in triplicate.

## 5. Conclusions

Recombinant proteins (rTLP and rCHI) were able to form haze and can be applied as haze-forming protein models for future research. rTLP and rCHI can be produced and purified by affinity tag chromatography, which is much faster than the tedious purification steps of HUPs from *V. vinifera*. The heterologously expressed proteins presented similar characteristics in terms of glycosylation, melting point, adsorption by bentonite and activity (CHI) in comparison to the native proteins. The exposure of thiol groups was evidenced under denaturation of rTLP and rCHI and might be associated with *S*-sulfonation reactions and rearrangement of disulfide bridges. Sulfite ions and polyphenols were confirmed as haze inducers, and polyphenols clearly participate in protein cross-linking reactions. The combination of the two different haze proteins increased the haze levels, ensuring that haze in wines is caused by the aggregation of several proteins from the grape plant and also possibly from yeast.

## Figures and Tables

**Figure 1 molecules-27-06409-f001:**
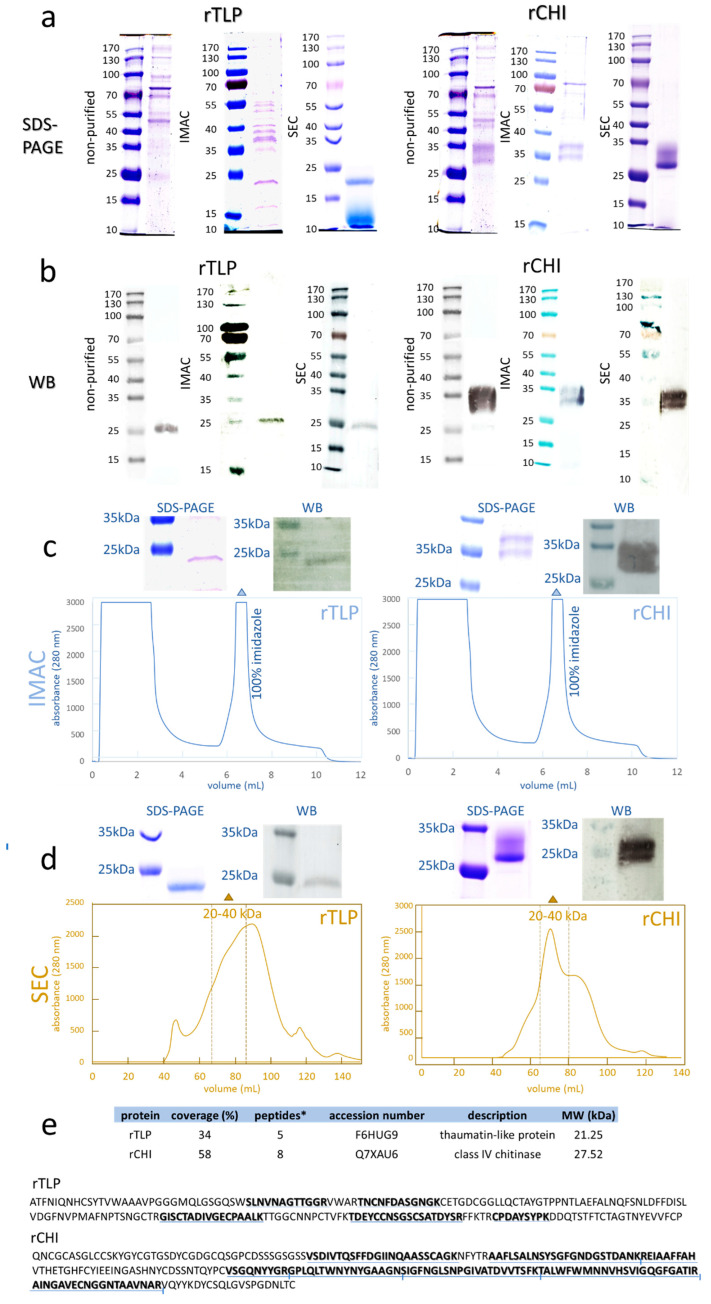
Analysis of recombinant TLP (rTLP) and CHI (rCHI). (**a**) Electrophoresis gels (SDS-PAGE) show the separation of the non-purified protein fractions and the protein fractions purified by IMAC and SEC. (**b**) WB membranes showing the detected his-tagged rTLP and rCHI in the non-purified fractions and the protein fractions purified by IMAC and SEC. (**c**) IMAC chromatograms of the fermentation extracts (selected clones) of rTLP and rCHI. (**d**) SEC chromatogram of the eluted peak (100% imidazole) of rTLP and rCHI (in Figure 1c). (**e**) Protein identification based on MS analysis of tryptic peptides and their identification (underlined in blue) in the amino acid sequences of rTLP and rCHI. * Means unique peptides.

**Figure 2 molecules-27-06409-f002:**
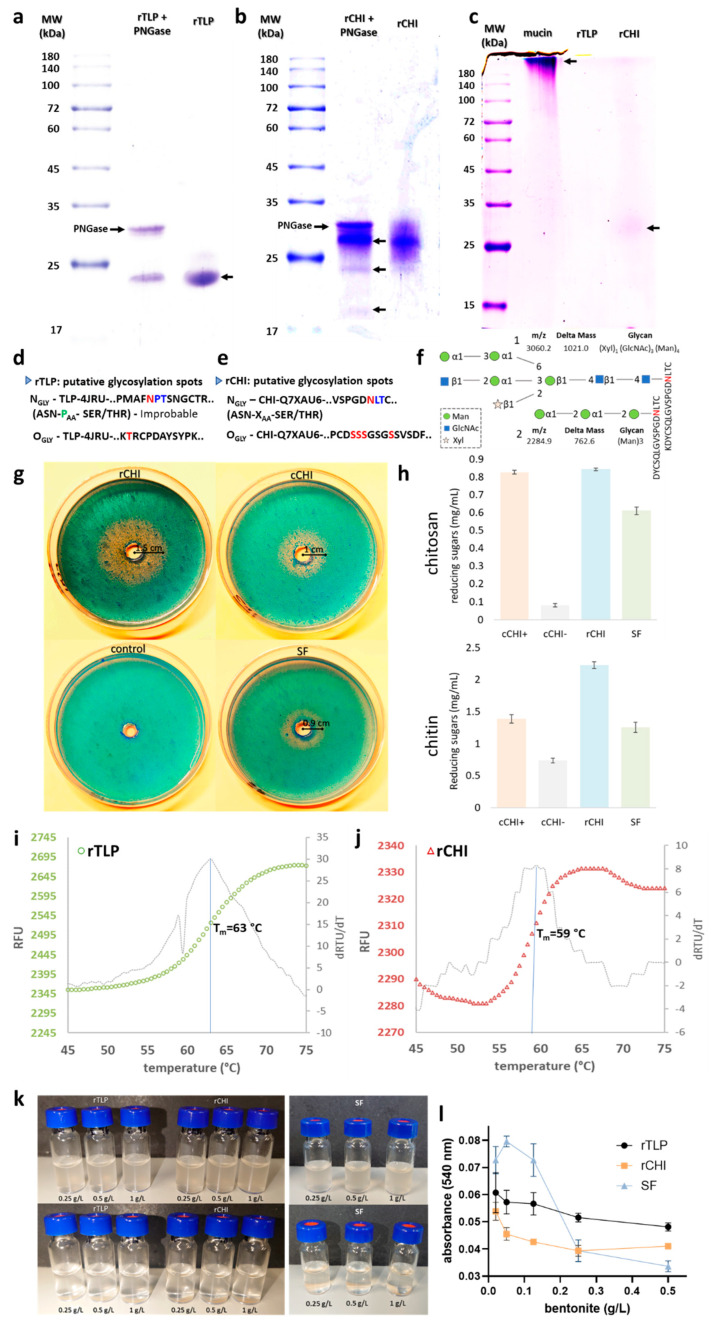
Characterization of rTLP and rCHI in terms of glycosylation, activity (CHI), melting point and adsorption by bentonite. The glycosylation analysis of rTLP and rCHI by (**a**) comparison of PNGase treated and non-treated rTLP. (**b**) Comparison of PNGase treated and non-treated rCHI (protein bands are indicated with arrows). (**c**) Identification of glycoproteins by the Schiff-reagent method after SDS-PAGE (the highly glycosylated mucin protein was used as control and protein bands are indicated with arrows). The predicted glycosylations are presented in (**d**,**e**) by showing the putative glycosylation sites for rTLP and rCHI, respectively. (**f**) two glycans (numbered 1 and 2) identified by MS-based analysis were found attached to the Asn (N) residue (highlighted in red) of rCHI. The mass-to-charge ratio (*m*/*z*) and the differences in the peptide masses (due the glycan attachment) are also shown (**g**) Chitinolytic activity of rCHI assessed by the agar diffusion method and stained by the calcofluor white stain reagent. A commercial chitinase from *Streptomyces griseus* (cCHI) was used as positive control. A pre-heated chitinase (denatured cCHI, used as negative control) and chitinases present in proteins from the Silvaner Franken wine (SF). (**h**) DNS assays with rCHI, cCHI, cCHI-(denatured) and SF using two chitinous substrates (chitin and chitosan). CPM fluorescence signal of rTLP and of rCHI are shown in (**i**,**j**) with their respective first derivatives. (**k**) Buffered solutions (pH 4) of rTLP, rCHI and SF (on the right) with addition of bentonite pre (upper part) and post (lower part) a heat test. (**l**) Quantitative analysis of the adsorption of the protein (rTLP, rCHI and SF) to bentonite with haze threshold concentrations (0.02 to 0.5 g/L).

**Figure 3 molecules-27-06409-f003:**
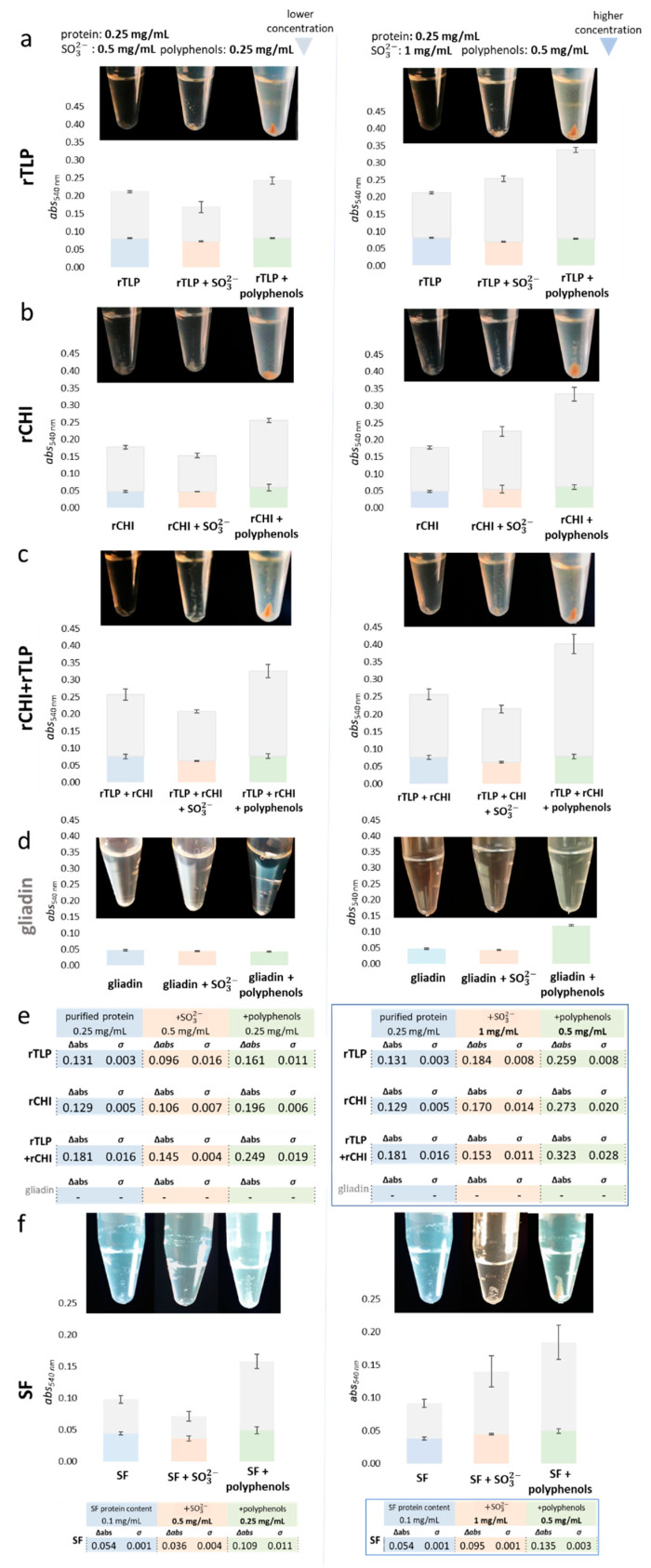
Influence of ${\mathrm{SO}}_{3}^{2-}$ and polyphenols on the protein aggregation and haze levels of rCHI, rTLP and controls. Aggregation levels are presented in terms of absorbance (in bars) and the visible residual pellet formed (at the bottom of micro tubes); the experimental variants are divided into purified protein, supplied with sulfite (+$\;{\mathrm{SO}}_{3}^{2-}$), and supplied with polyphenols. The gray segment of each bar represents the absorbance after the heat test (haze formation) and the colored part represents the final absorbance after centrifugation. The columns on the left and right display experiments with lower and higher concentrations of matrix compounds or additive (sulfite and polyphenols), respectively. The letters on the horizontal axis show the results for different proteins: (**a**) TLP, (**b**) CHI, (**c**) CHI+TLP and (**d**) gliadin (used as control). (**e**) The tables show the absorbance values of formed haze (at 540 nm) under the different experimental conditions and the highlighted table (outlined by blue square) shows the absorbance values (540 nm) at higher concentrations of sulfite and polyphenols. Haze levels of protein solutions from the SF wine under the same experimental conditions, are shown in (**f**).

**Figure 4 molecules-27-06409-f004:**
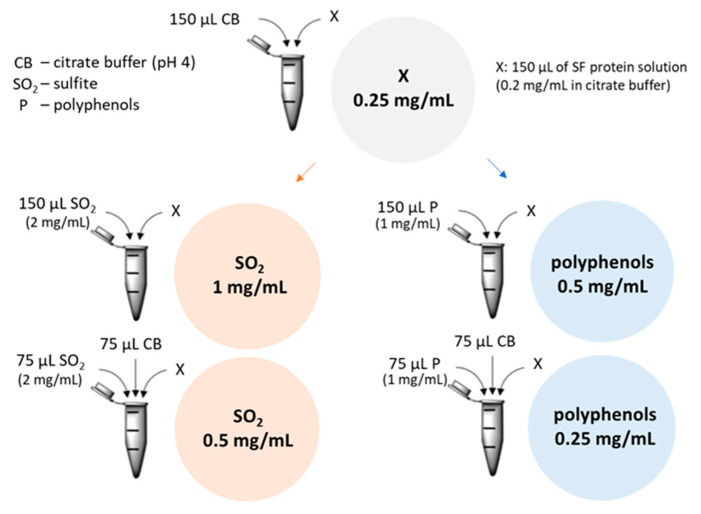
Recombinant TLP and CHI at different concentrations with sulfite ions and polyphenols at different concentrations were used to measure their influence on the haze potential. CB: 0.1 M citrate buffer; ${\mathrm{SO}}_{3}^{2-}$: Sulfite ion (from Na_2_SO_3_ solution) (2 mg/mL); P: Polyphenol extract solution (2 mg/mL); x: 150 µL of protein solution (rTLP, rCHI or gliadin at 0.5 mg/mL) or 75 µL of TLP and 75 µL of CHI solutions in case of combination.

## Data Availability

The raw MS data related to the recombinant rTLP and rCHI are deposited to the ProteomeXchange Consortium via the PRIDE partner repository with the dataset identifier PXD035796. Additional data related to this paper may be requested from the authors.

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
