# Peer review of "Recombinant Thaumatin-Like Protein (rTLP) and Chitinase (rCHI) from *Vitis vinifera* as Models for Wine Haze Formation"

_molecules, 2022, doi:10.3390/molecules27196409_

Round 1

Reviewer 1 Report

In this manuscript the authors proposed the use of TLP (rTLP) and CHI (rCHI), expressed by Komagataella phaffii, as haze-protein models. The authors claimed that the proteins can be applied to study haze formation mechanisms on a molecular level and to explore alternative fining methods by screening proteolytic enzymes and ideal adsorptive resins.

The work is well done, the manuscript is well written, graphics are very descriptive and clear, and the results are interesting, they will probably be useful for biotechnological purposes.

There are only minor issues that need to be corrected.

·        First, genes and scientific names should be italicized.

·        Second, add the economic importance of this study in the introduction.

·        Finally, increase the resolutions for the figures in figure 2 (especially figure h)

Author Response

There are only minor issues that need to be corrected.

  • First, genes and scientific names should be italicized.

Answer: All the genes and scientific species names have been italicized.

  • Second, add the economic importance of this study in the introduction.

Answer: As suggested, a sentence was added to emphasize the economic impact of the present study in lines 66-68:

“The use of rCHI and rTLP may be crucial for the search for proteolytic enzymes and adsorptive agents as alternative fining agents, providing the opportunity to find suitable and profitable alternatives for winemakers”.

  • Finally, increase the resolutions for the figures in figure 2 (especially figure h)

Answer: As requested, the quality, size and text of the figures were improved. The updated figures were added to the main manuscript (particularly Figure 2h was edited).

Reviewer 2 Report

This article propose the usage of recombinant TLP (rTLP) and CHI (rCHI) as haze-protein models. The topic is relevant and well-structured but few issues need to be addressed:

-The introduction part need to be change and improved. The description of your objective should be shorter and the current state of the research field should be reviewed carefully.

- Escherichia coli and  K. phaffii : must be writen in italic

Author Response

-The introduction part need to be change and improved. The description of your objective should be shorter and the current state of the research field should be reviewed carefully.

Answer: The introduction was modified and a better description of the experiments was added:

“We propose the comparison of recombinant TLPs and CHIs (rTLP and rCHI) to serve as alternative models of wine haze proteins. Therefore, these proteins were heterologously expressed by Komagataella phaffii and compared with their corresponding native-host proteins in terms of glycosylation, melting point, aggregation potential, adsorption by bentonite and functionality (CHI activity).”

Instead of:

“We proposed the comparison of authentic and recombinant TLPs and CHIs (rTLP and rCHI) to serve as alternative models of wine haze proteins. Therefore, those proteins were expressed proteins heterologously using the methylotrophic yeast Komagataella phaffii. Next, the expression of them was validated by mass spectrometry (MS)-based proteomics and western blot (WB) analysis, and also compared them with their corresponding native-host proteins in terms of glycosylation, melting point, aggregation potential, adsorption by bentonite and functionality (CHI activity).”

One sentence was added to emphasize the economic impact of the present study:

“The use of rCHI and rTLP may be crucial for the search for proteolytic enzymes and adsorptive agents as alternative fining agents, providing the opportunity to find suitable and profitable alternatives for winemakers”.

Section 3.1 (discussion) has been revised.

- Escherichia coli and K. phaffii: must be writen in italic

Answer: All genes and scientific names have been italicized.